# Effect of *Cardamine violifolia* on Plasma Biochemical Parameters, Anti-Oxidative Capacity, Intestinal Morphology, and Meat Quality of Broilers Challenged with Lipopolysaccharide

**DOI:** 10.3390/ani12192497

**Published:** 2022-09-20

**Authors:** Yu Wei, Qingyu Gao, Xiaoqing Jing, Yue Zhang, Huiling Zhu, Xin Cong, Shuiyuan Cheng, Yulan Liu, Xiao Xu

**Affiliations:** 1Hubei Key Laboratory of Animal Nutrition and Feed Science, National R&D Center for Se-Rich Agricultural Products Processing, Wuhan Polytechnic University, Wuhan 430023, China; 2Enshi Se-Run Material Engineering Technology Co., Ltd., Enshi 445000, China

**Keywords:** *Cardamine violifolia*, broilers, selenium, antioxidant, meat quality, LPS

## Abstract

**Simple Summary:**

Selenium (Se) is a crucial trace nutrient for animal maintenance and growth as well as human health. Organic Se has less poisonous and better tissue retention as well as bioavailability compared to inorganic Se. *Cardamine violifolia* is a newly discovered Se-enriched plant rich in MeSeCys and SeCys and has a strong antioxidant capacity. This study aimed to investigate the effects of *Cardamine violifolia* on plasma biochemical indices, antioxidant levels, intestinal morphology, and meat quality of broilers under acute LPS-induced oxidative stress by comparing it with inorganic Se (sodium selenite). The results showed that *Cardamine violifolia* alleviated tissue injury, enhanced antioxidant capacity, and improved meat quality of breast and thigh muscle after LPS stress.

**Abstract:**

*Cardamine violifolia* is a newly discovered selenium (Se)-enriched plant rich in MeSeCys and SeCys and has a strong antioxidant capacity. This study aimed to investigate the effects of *Cardamine violifolia* on plasma biochemical indices, antioxidant levels, intestinal morphology, and meat quality of broilers under acute LPS-induced oxidative stress by comparing it with inorganic Se (sodaium selenite). A total of 240 one-day-old Ross 308 broilers were fed a basal diet and divided into four groups: (1) SeNa-SS, fed a diet supplied with 0.3 mg/kg Se from sodium selenite, and injected with 0.9% sterile saline, (2) SeCv-SS, fed a diet supplied with 0.3 mg/kg Se from *Cardamine violifolia*, and injected with 0.9% sterile saline, (3) SeNa-LPS, fed a diet supplied with 0.3 mg/kg Se from sodium selenite, and injected with 0.5 mg/kg LPS, (4) SeCv-LPS, fed a diet supplied with 0.3 mg/kg Se from *Cardamine violifolia* and injected with 0.5 mg/kg LPS. The experiment lasted for 42 days. Sterile saline or LPS was injected intraperitoneally two hours before slaughter, and blood and tissue samples were collected for testing. The results showed that compared with SeNa, SeCv significantly reduced the plasma levels of aspartate aminotransferase, alanine aminotransferase, and urea nitrogen after LPS challenge (*p* < 0.05), and increased the plasma levels of total antioxidant capacity and glutathione peroxidase, decreased malondialdehyde content in LPS-challenged broilers (*p* < 0.05). In addition, compared with SeNa, SeCv supplementation increased villus height and the ratio of villus height to crypt depth of jejunum and ileum after LPS challenge (*p* < 0.05). Additionally, SeCv could increase the redness of breast and thigh muscle, and decrease drip loss, cooking loss, and shear force (*p* < 0.05). In conclusion, our results indicated that supplementing with 0.3 mg/kg Se from *Cardamine violifolia* alleviated tissue injury after LPS challenge, increased antioxidant capacity, and improved meat quality of breast and thigh muscle after stress.

## 1. Introduction

Selenium (Se) is a crucial trace nutrient for animal maintenance and growth as well as human health [1]. Many studies have demonstrated that Se supplementation in poultry diets plays an essential role in maintaining poultry health and production [2]. Se is an important component of at least 25 selenoproteins involved in various physiological processes, including immune function and maintenance of antioxidant defense to avoid tissue damage [3]. Among them, the most important role of Se lies in that as a key component of antioxidant enzymes, which enhances the activity of antioxidant enzymes, such as glutathione peroxidase (GSH-Px), superoxide dismutase (SOD), and total antioxidant capacity (T-AOC), and reduces the content of lipid peroxide malondialdehyde (MDA), and reduces oxidative stress in broilers [4,5]. In addition, dietary Se supplementation can also improve the growth performance and meat quality of broilers [6,7,8].

Se is mostly supplied to poultry diets as inorganic and organic Se. Sodium selenite (SeNa), the most widely utilized inorganic form of Se, has a fairly narrow range between dietary requirements and toxicity [9]. Se-enriched yeast is an organic Se that is frequently used. Organic Se is less poisonous than inorganic Se and has better tissue retention and bioavailability than inorganic Se, which may be due to the absorption of organic Se in the gut being faster and higher [10]. In addition, different absorption pathways will result in organic Se being more retained in muscle tissue than inorganic Se [5]. Therefore, organic Se is preferable in poultry farming. *Cardamine violifolia*, a new super-selenium-rich plant, was recently discovered in Enshi, Hubei Province, China. Studies have shown that the Se content in its leaves exceeds 1400 mg/kg (dry weight), of which about 14% is in the form of the water-soluble protein [11]. Studies have found that its Se mainly exists in the form of organic Se, and the content of organic Se in protein is much higher than that in Se-enriched yeast, Se-enriched peanuts, and other Se-enriched products [12]. Its organic Se forms are mainly MeSeCys and SeCys, which have strong antioxidant capacity [13].

Although many studies have shown that organic Se could increase Se content in animal products, improve meat quality, and enhance antioxidant capacity [14,15], it is not clear whether *Cardamine violifolia* can be used as an effective Se source for the production of broiler industry. Therefore, this study constructed a stress model by intraperitoneal injection of LPS in broilers to explore the potential of *Cardamine violifolia* as a new Se source additive to improve plasma biochemical parameters, antioxidant status, intestinal health, and meat quality of broilers stimulated by LPS [16].

## 2. Materials and Methods

### 2.1. Experimental Animals and Design

The animal trial was conducted according to the Animal Scientific Procedures Act 1986 (Home Office Code of Practice. HMSO: London January 1997) and EU regulations (Directive 2010/63/EU). This experimental protocol (No. WPU202112056) used in this study was approved by the Institutional Animal Care and Use Committee of Wuhan Polytechnic University (Wuhan, China). A total of 240 one-day-old Ross 308 male broilers were randomly divided into 4 groups with 6 replicates per group and 10 broilers per replicate according to a 2 × 2 factorial design. The four treatments were as follows: (1) SeNa-SS group, supplemented with 0.3 mg/kg Se from sodium selenite and injected with sterile saline, (2) SeCv-SS group, supplemented with 0.3 mg/kg Se from *Cardamine violifolia* and injected with sterile saline, (3) SeNa-LPS group, supplemented with 0.3 mg/kg Se from sodium selenite and injected with LPS, and (4) SeCv-LPS group, supplemented with 0.3 mg/kg Se from *Cardamine violifolia* and injected with LPS. At 42 days of age, one chicken in each replicate in SeNa-LPS and SeCv-LPS groups was intraperitoneally injected with LPS at 0.5 mg/kg BW, and one chicken in each replicate in SeNa-SS and SeCv-SS groups was intraperitoneally injected with 0.9% sterile saline. All chickens were kept in wire cages (90 × 60 × 40 cm^3^) with free access to pelleted feed and water and constant light, averaging about 16–18 h a day. The ambient temperature was maintained at 35 to 37 degrees Celsius for the first week, then gradually decreased at a rate of about 3 degrees Celsius per week until a final temperature of 21 degrees Celsius was reached, with relative humidity controlled at 55–65%. *Cardamine violifolia* used in this study was provided by Enshi Se-Run Material Engineering Technology Co., Ltd., Enshi, China. This *Cardamine violifolia* included leaves and stalk with 1430 mg Se per kg dry weight mainly existed as the forms of SeCys and MeSeCys. The sodium selenite and selenium yeast samples were commercial products purchased from Angel Yeast Co., Ltd., Yichang, China. LPS (serotype O55:B5) was provided by Sigma Chemical Ltd. Co. St. Louis, MO, USA.

### 2.2. Sample Collection

Two hours after LPS injection, blood was collected by sub wing vein and centrifuged at 3500 rmp for 10 min. Plasma was extracted and stored at −80 °C for analysis. After blood collection, one chicken was selected from each replicate, and euthanasia was performed by the cervical spine dislocation method, left breast muscle and thigh muscle were collected, duodenum, jejunum, and ileum were collected, and 2–3 cm of tissue was cut in the middle position and fixed in 10% formaldehyde-phosphate buffer for morphological evaluation.

### 2.3. Plasma Biochemical Indicators Measurement

Plasma biochemical parameters were determined using an automatic biochemical analyzer (7100, HITACHI, Tokyo, Japan), as previously described [17]. These included total protein (TP), albumin (ALB), glucose (GLU), triglyceride (TG), aspartate aminotransferase (AST), alanine aminotransferase (ALT), urea nitrogen (BUN).

### 2.4. Plasma Anti-Oxidative Capacity Measurement

T-AOC, GSH-Px, SOD enzyme activity, and MDA content in plasma were determined using the kit of Nanjing Jiancheng Institute of Biological Engineering (Nanjing, China) according to the manufacturer’s instructions. The activities of GSH-Px and SOD were expressed as units (U) per milliliter; the activity of T-AOC was expressed as mM, and the plasma content of MDA was expressed as nanomoles (nmol) per milliliter.

### 2.5. Intestinal Morphology Measurement

For intestinal morphology, duodenum, jejunum, and ileum were collected and drained and washed, and the intestinal segment with a position of about 2 cm in the middle was fixed in 10% formaldehyde–phosphate buffer, then embedded in paraffin, and stained with hematoxylin and eosin for histomorphological analysis. Villus height (VH), crypt depth (CD), and the ratio of villus height to crypt depth (VH/CD) were measured under a 100 × magnification optical microscope (Olympus CX31, Japan) and computer-aided morphometry system (BioScan Optimetric; BioScan Inc., Edmond, WA, USA), as previously described [18]. VH was measured from the junction of the villus tip to the crypt, and CD was measured from the junction of the villus crypt to the tip of the mucous muscularis [19].

### 2.6. Meat Quality of the Breast and Thigh Muscle Measurement

Meat quality indicators include meat color, drip loss, cooking loss, and shear force. The color of flesh was measured by colorimeter (CR-410, Konica Minolta, Tokyo, Japan) for lightness (*L**), redness (*a**), and yellowness (*b**) of the three same positions (middle, medial, and lateral) of the breast and thigh muscle [20]. The drip loss of raw meat for 24 h after storage was measured by the plastic bag method: about 30 g of muscle was obtained and hung in a plastic bag, stored at 4 °C for 24 h, then wiped with absorbent paper and weighed again. The difference in weight corresponded to the drip loss and was expressed as a percentage of the initial muscle weight [21]. Based on previous studies, muscle cooking loss was measured by the cooking method. In simple terms, approximately 30 g of muscle is weighted after 20 min in an 80 °C water bath and then weighed again. The weight difference corresponds to cooking loss and was expressed as a percentage of initial muscle weight [22]. Following previous research, muscle shear force was measured by cutting 3 cm × 1 cm × 1 cm shapes perpendicular to the fiber direction of the muscle after the cooked muscle was cooled to room temperature and determined using the Instron Universal Mechanical Machine (TA500 Lloyd Texture Analyzer fitted with a triangular Warner-Bratzler shear, Lloyd instruments, Bognor Regis, UK) [23].

### 2.7. Statistical Analyses

The data were analyzed by ANOVA using the general linear model (GLM) procedure of SAS 9.1 as a 2 × 2 factorial arrangement with diet and LPS as the main effects. An individual bird was the experimental unit for the parameters. When the *p* value of the interaction of main effects was less than 0.05, differences among the treatments were examined by ANOVA using Duncan’s multiple range test, and differences between means at *p* < 0.05 were considered to be statistically significant. All data were expressed as treatment means with their pooled SEM.

## 3. Results

### 3.1. Plasma Biochemical Indicators

The effect of *Cardamine violifolia* on plasma biochemical indicators of broilers challenged and unchallenged intraperitoneally with LPS is shown in Table 1. There was a significant interaction between the LPS challenge and SeCv supplementation for BUN, AST, and ALT (*p* < 0.05). Supplementation with SeCv significantly decreased the content of BUN and the activity of AST and ALT in the broilers challenged with LPS (*p* < 0.05). Moreover, the broilers fed SeCv diet had significantly decreased plasma TG levels (*p* < 0.05).

### 3.2. Plasma Anti-Oxidative Capacity

The effect of *Cardamine violifolia* on plasma anti-oxidative capacity of broilers challenged and unchallenged intraperitoneally with LPS is shown in Table 2. There was a significant interaction between LPS challenged and SeCv supplementation for the enzyme activity of T-AOC and GSH-Px and the concentration of MDA (*p* < 0.05). Compared to supplementation with SeNa, supplementation with SeCv significantly enhanced the activities of T-AOC and GSH-Px in the broilers challenged with LPS (*p* < 0.05). Moreover, supplementation with SeCv significantly reduced MDA content in the broilers challenged with LPS (*p* < 0.05).

### 3.3. Intestinal Morphology

The effect of *Cardamine violifolia* on the intestinal morphology of broilers challenged and unchallenged intraperitoneally with LPS is shown in Table 3. There was a significant interaction between LPS challenge and SeCv supplementation for VH of the duodenum, jejunum, and ileum, and the ratio of VH/CD of jejunum and ileum (*p* < 0.05). Compared to the broiler treated with saline, the VH of the duodenum, jejunum, and ileum was significantly lower in the broiler treated with LPS. Compared to supplementation with SeNa, supplementation with SeCv led to enhanced VH in the broiler challenged with LPS (*p* < 0.05). Meanwhile, compared with supplementation of SeNa, higher VH/CD of jejunum and ileum was observed in the broiler injected with LPS fed with SeCv (*p* < 0.05). Similar to the above results, the histological appearance showed that supplementation of SeCv reduced LPS-induced intestinal mucosal injury in broilers compared with feeding SeNa (Figure 1). Obvious reductions in villus height and intestinal mucosal injury were obtained in the broilers injected with LPS fed with SeNa. However, these intestinal injuries induced by LPS were alleviated in the broilers injected with LPS fed with SeCv.

### 3.4. Meat Quality of the Breast and Thigh Muscle

The effect of *Cardamine violifolia* on the meat quality of breast and thigh muscle of broilers challenged and unchallenged intraperitoneally with LPS is shown in Table 4. There was a significant interaction between LPS challenge and SeCv supplementation for redness (*a**), drip loss, cooking loss, and shear force of the breast and thigh muscle. Compared with supplementation of SeNa, redness (*a**) of the breast and thigh muscle was significantly enhanced in the broiler challenged with LPS fed with SeCv (*p* < 0.05). In addition, supplementation with SeCv significantly decreased drip loss, cooking loss, and shear force of the breast and thigh muscle in the broiler challenged with LPS (*p* < 0.05).

## 4. Discussion

This study aimed to investigate whether *Cardamine violifolia* could be used as a novel Se-enriched feed additive to alleviate LPS-induced stress in broilers. Compared with SeNa supplementation, *Cardamine violifolia* supplementation significantly enhanced anti-oxidative capability and liver health status. Moreover, *Cardamine violifolia* supplementation effectively alleviated intestinal injury induced by LPS and improved the meat quality of the breast and thigh muscle in broilers.

Blood is a key indicator of health, and specific factors such as nutrition, disease, and environment can lead to changes in blood biochemistry [24]. In plasma biochemical analysis, the content of enzyme activities, such as AST, ALT, and metabolite BUN, are related to the health of the liver and kidney [25]. The activity of these enzymes and the level of metabolites are related to stress indicators [26,27,28], associated with antioxidant status and liver and kidney damage. In this study, the levels of AST, ALT, and BUN were significantly increased in broilers injected with LPS. However, compared with SeNa, the levels of AST, ALT, and BUN were significantly decreased when fed the *Cardamine violifolia* diets. The results showed that the stress response of broilers challenged with LPS resulted in impaired metabolism of the liver and kidney [29,30,31]. Feeding *Cardamine violifolia* could significantly alleviate the oxidative stress response induced by LPS and reduce the levels of AST, ALT, and BUN after stress. Moreover, a large volume of literature has shown that dietary supplementation with organic Se can reduce serum levels of AST, ALT, and BUN [30,32]. The main reason may be that Se protects the body from oxidative damage by regulating redox reactions. As a plant Se source rich in MeSeCys and SeCys, *Cardamine violifolia* has a higher bioavailability than SeNa.

LPS can induce an oxidative stress response and reduce the antioxidant capacity of broilers [33,34]. T-AOC is the total antioxidant level composed of various antioxidant substances and antioxidant enzymes, reflecting the cumulative effect of antioxidants in the body [35]. MDA is an important marker reflecting the degree of lipid peroxidation and cell damage [36]. GSH-Px is an important peroxide-decomposition enzyme in the body, which reduces toxic peroxides to non-toxic hydroxyl compounds and promotes the decomposition of H_2_O_2_, thus protecting the structure and function of cell membranes from the interference and damage of oxides [37]. In this study, the levels of T-AOC and GSH-Px in plasma were significantly decreased, and the content of MDA was significantly increased by injecting LPS. However, the *Cardamine violifolia* supplementation increased plasma T-AOC and GSH-Px levels and decreased MDA content in LPS-challenged broilers compared with the SeNa supplementation. The experimental results showed that LPS injection induced oxidative stress in the body, leading to the decrease of antioxidant levels, and feeding *Cardamine violifolia* could effectively enhance antioxidant capacity. Many studies have reported that *Cardamine violifolia* has a strong antioxidant capacity for the reason that selenium is the component of many antioxidant enzymes such as GSH-Px [11,38]. Consistent with our findings, it was demonstrated that dietary *Cardamine violifolia* supplementation increased serum T-AOC and GSH-Px levels and decreased MDA content in high-fat diet-induced obese mice [39]. Compared with SeNa, *Cardamine violifolia* can effectively alleviate oxidative stress induced by LPS, which may be attributed to its herb origin, low toxicity, and higher bioavailability.

Intestinal morphology is an essential indicator reflecting the health status of the digestive tract [40]. VH and CD are closely related to nutrient digestion and absorption by VH/CD [41,42]. LPS can induce oxidative stress and lead to intestinal morphology damage [43,44]. Compared with SeNa, *Cardamine violifolia* supplementation increased VH in the small intestine (duodenum, jejunum, and ileum) and VH/CD in jejunum and ileum after being LPS challenged. A large number of studies have shown that dietary supplementation of organic selenium can increase intestinal VH compared with inorganic selenium [45], and increase VH/CD [46]. In addition, studies have shown that organic Se is highly associated with larger intestinal villi in virus-infected broilers compared to sodium selenite [47]. As a novel Se source rich in MeSeCys and SeCys, *Cardamine violifolia* exhibits the same properties as organic Se and may act as an exogenous antioxidant factor to positively affect intestinal cell vitality and thus improve intestinal morphology under stress conditions.

Meat color, drip loss, cooking loss, and shear force are the main indicators of meat quality. Color is crucial to the appearance of meat and consumer preference [48]. Drip loss, cooking loss, and the ability of shear force reaction to retain water not only affect the storage quality and processing benefit of meat but also affect the tenderness of meat [49]. Studies have shown that pre-slaughter stress response has a significant impact on meat quality [50,51]. The results of this study showed that LPS injection before slaughter induced oxidative stress, resulting in decreased meat redness value, drip loss and cooking loss, and increased shear force. Compared with SeNa, *Cardamine violifolia* increased chest and leg muscle redness (*a**), decreased drip loss, cooking loss, and shear force after being LPS challenged. Consistent with our results, compared with inorganic selenium, adding organic selenium can improve the redness value of breast muscle and leg muscle of broilers [52], reduce drip loss and cooking loss [53], and reduce shear force and increase tenderness [54]. The results may be attributed to the fact that *Cardamine violifolia*, as a novel Se source rich in MeSeCys and SeCys, plays the same role as organic Se in effectively preventing the oxidation of myoglobin or oxymyoglobin to ferric metmyoglobin, thereby improving meat quality and maintaining cell integrity to reduce drip loss and cooking loss. Moreover, the reduction of shear force may be due to the higher bioavailability of selenium in *Cardamine violifolia*.

## 5. Conclusions

Diet supplied with 0.3 mg/kg Se from *Cardamine violifolia*, a novel Se source rich in MeSeCys and SeCys, significantly reduced LPS-induced tissue injury, and increased plasma antioxidant levels compared with SeNa. In addition, *Cardamine violifolia* alleviated LPS-induced intestinal injury and improved the meat quality of breast and thigh muscles under LPS-induced stress.

## Figures and Tables

**Figure 1 animals-12-02497-f001:**
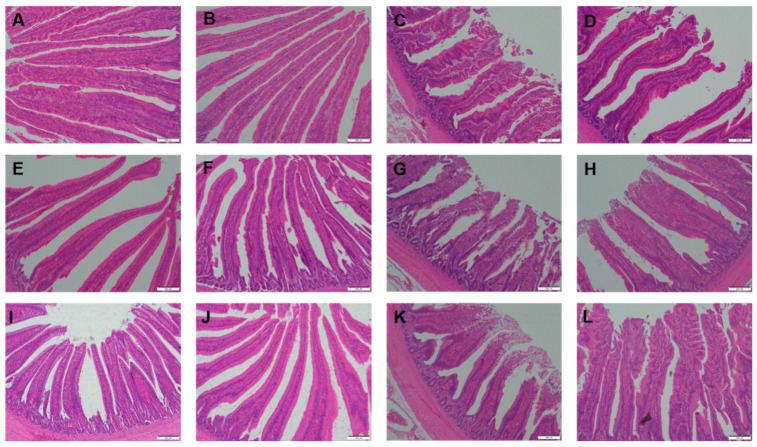
Intestinal mucosal histological appearance (hematoxylin and eosin) of the effect of *Cardamine violifolia* on the intestinal morphology of broilers challenged and unchallenged intraperitoneally with LPS. Original magnification 100×. Scale bars = 200 μm. (**A**–**D**) Duodenum histological appearance of the broilers treated by SeNa-SS, SeCv-SS, SeNa-LPS, and SeCv-LPS, respectively. (**E**–**H**) Jejunum histological appearance of the broilers treated by SeNa-SS, SeCv-SS, SeNa-LPS, and SeCv-LPS, respectively. (**I**–**L**) Ileum histological appearance of the broilers SeNa-SS, SeCv-SS, SeNa-LPS, and SeCv-LPS, respectively.

**Table 1 animals-12-02497-t001:** Effect of *Cardamine violifolia* on plasma biochemical indicators of broilers challenged and unchallenged intraperitoneally with LPS.

Item	Saline		LPS		SEM	*p*-Value		
	SeNa	SeCv	SeNa	SeCv		Diets	LPS	Interaction
TP, g/L	23.5	24.5	22.4	22.7	2.01	0.358	0.414	0.798
ALB, g/L	12.5	12.7	12.4	12.5	1.11	0.845	0.715	0.920
GLU, mmol/L	10.3	10.4	10.5	10.6	0.35	0.685	0.465	0.882
TG, mmol/L	2.86	2.34	2.78	2.38	0.17	0.021	0.835	0.924
BUN, mmol/L	0.51 ^b^	0.42 ^b^	0.62 ^a^	0.44 ^b^	0.04	<0.001	0.273	0.014
AST, U/L	237 ^b^	220 ^b^	295 ^a^	266 ^ab^	20	0.024	0.034	0.038
ALT, U/L	4.50 ^b^	4.48 ^b^	5.34 ^a^	4.62 ^b^	0.28	0.358	0.260	<0.001

N = 6 (1 bird per cage). The same letter on the shoulder of the mean in the same line indicates that the difference is insignificant, and the absence of the same letter means that the difference is significant. TP, total protein; ALB, albumin; GLU, glucose; TG, total triglyceride; BUN, blood urea nitrogen; AST, aspartate aminotransferase; ALT, alanine aminotransferase; SEM, standard error of the mean.

**Table 2 animals-12-02497-t002:** Effect of *Cardamine violifolia* on plasma anti-oxidative capacity of broilers challenged and unchallenged intraperitoneally with LPS.

Item	Saline		LPS		SEM	*p*-Value		
	SeNa	SeCv	SeNa	SeCv		Diets	LPS	Interaction
T-AOC, mM	1.15 ^a^	1.29 ^a^	0.87 ^b^	1.12 ^a^	0.11	0.320	0.129	<0.001
GSH-Px, U/mL	1423 ^a^	1527 ^a^	1017 ^b^	1458 ^a^	126	<0.001	0.256	<0.001
SOD, U/mL	298	302	223	235	24	0.748	<0.001	0.820
MDA, nmol/mL	2.40 ^b^	2.23 ^b^	2.87 ^a^	2.43 ^b^	0.18	0.245	0.028	<0.001

N = 6 (1 bird per cage). The same letter on the shoulder of the mean in the same line indicates that the difference is insignificant, and the absence of the same letter means that the difference is significant. T-AOC, total antioxidant capacity; GSH-Px, glutathione peroxidases; SOD, superoxide dismutase; MDA, malondialdehyde; SEM, standard error of the mean.

**Table 3 animals-12-02497-t003:** Effect of *Cardamine violifolia* on the intestinal morphology of broilers challenged and unchallenged intraperitoneally with LPS.

Item	Saline		LPS		SEM	*p*-Value		
	SeNa	SeCv	SeNa	SeCv		Diets	LPS	Interaction
Duodenum								
VH, μm	985 ^a^	1025 ^a^	804 ^b^	945 ^a^	35	0.030	0.011	<0.001
CD, μm	231	238	220	241	17	0.429	0.893	0.660
VH/CD	4.26	4.31	3.65	3.92	0.14	0.253	<0.001	0.088
Jejunum								
VH, μm	1001 ^a^	1038 ^a^	788 ^c^	895 ^b^	34	0.313	<0.001	<0.001
CD, μm	227	220	245	220	15	0.860	0.799	0.931
VH/CD	4.41 ^ab^	4.72 ^a^	3.22 ^c^	4.07 ^b^	0.15	<0.001	<0.001	<0.001
Ileum								
VH, μm	810 ^a^	826 ^a^	711 ^b^	789 ^a^	26	0.081	0.043	<0.001
CD, μm	201	196	204	200	13	0.741	0.960	0.729
VH/CD	4.03 ^a^	4.21 ^a^	3.49 ^b^	3.95 ^a^	0.14	0.014	<0.001	<0.001

N = 6 (1 bird per cage). The same letter on the shoulder of the mean in the same line indicates that the difference is insignificant, and the absence of the same letter means that the difference is significant. VH, villus height; CD, crypt depth; VH/CD, the ratio of villus height to crypt depth; SEM, standard error of the mean.

**Table 4 animals-12-02497-t004:** Effect of *Cardamine violifolia* on the meat quality of breast and thigh muscle of broilers challenged and unchallenged intraperitoneally with LPS.

Item	Saline		LPS		SEM	*p*-Value		
	SeNa	SeCv	SeNa	SeCv		Diets	LPS	Interaction
Breast muscle								
Color								
* L**	58.4	53.6	57.6	58.0	2.88	0.775	0.690	0.489
*a**	14.4 ^a^	14.9 ^a^	12.1 ^b^	14.2 ^a^	0.80	0.206	0.358	0.020
*b**	6.88	7.59	7.05	7.33	0.75	0.658	0.801	0.699
Drip loss, %	1.24 ^b^	1.13 ^b^	2.05 ^a^	1.36 ^b^	0.28	0.089	0.251	0.006
Cooking loss, %	26.4 ^b^	25.3 ^b^	31.1 ^a^	26.4 ^b^	1.31	0.236	0.331	0.014
Shear force, N	24.7 ^b^	25.1 ^b^	29.8 ^a^	25.4 ^b^	1.25	0.510	0.389	0.042
Thigh muscle								
Color								
*L**	60.3	62.4	63.5	61.6	2.98	0.818	0.798	0.843
*a**	18.9 ^a^	19.3 ^a^	14.8 ^b^	18.6 ^a^	1.16	0.013	0.189	<0.001
*b**	7.69	7.25	8.02	7.50	0.70	0.593	0.796	0.882
Drip loss, %	1.01 ^b^	0.89 ^b^	1.98 ^a^	1.12 ^b^	0.25	0.007	<0.001	<0.001
Cooking loss, %	22.6 ^b^	22.7 ^b^	28.8 ^a^	24.2 ^b^	1.20	0.425	<0.001	<0.001
Shear force, N	12.8 ^b^	11.6 ^b^	15.7 ^a^	12.2 ^b^	0.73	0.015	0.072	<0.001

N = 6 (1 bird per cage). The same letter on the shoulder of the mean in the same line indicates that the difference is insignificant, and the absence of the same letter means that the difference is significant. SEM, standard error of the mean.

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
