# Peer review of "Effect of Cardamine violifolia on Plasma Biochemical Parameters, Anti-Oxidative Capacity, Intestinal Morphology, and Meat Quality of Broilers Challenged with Lipopolysaccharide"

_animals, 2022, doi:10.3390/ani12192497_

Round 1

Reviewer 1 Report

The manuscript indicated that Cardamine violifolia, a novel Se-rich plant, have a positive effect on plasma parameter, oxidative stress, and intestinal morphology in LPS-challenged broilers. There are many literatures regarding to organic selenium, but the usage of the plant is the strong point of this study. The reviewer asks the following question and request some modification before considering the publication.

1.       The authors showed the supplementation of the plant improved oxidative stress and intestinal morphology in LPS-challenged broilers. In the discussion section, they state the plant has a strong antioxidant capacity. However, they should consider about how the organic selenium regulate the VH/CD ratio and the activity of GSH-Px.

2.       Generally, the organic selenium could improve the growth of broiler chickens. How about the growth performance or muscle weight in this study??

3.       Cardamine violifolia include organic selenium such as MeSeCys and SeCys. This study focused on organic selenium in the plant, however, the sulfur-containing amino acid also has an antioxidant capacity. The amino acids affect the positive effect of Cardamine violifolia in this study??

4.       L201-203; please state more detail about Figure 1.

5.       There is similar statement in section 2.3 and 2.6. These statement could merge??

6.       L104; Two hours later -> Two hours after LPS injection??

Author Response

Thank you for your review work. We have revised all the points according to your  comments.

Reviewer 2 Report

animals-1927265

Effect of Cardamine violifolia, a Se-enriched plant, on plasma biochemical parameters, anti-oxidative capacity, intestinal morphology, and meat quality of broilers challenged with lipopolysaccharide

1.  Wei et al.  Investigated the impact of the Cardamine violifolia on plasma biochemical indices, antioxidant levels, intestinal morphology, and meat quality of broilers under acute LPS-induced oxidative stress by comparing it with inorganic Se (sodium selenite). The results showed that Cardamine violifolia alleviated tissue injury, enhanced antioxidant capacity, and improved meat quality of breast and thigh muscle after LPS  stress.  The Ms  discussed important topic and had good scientific and practical impact of producing improved meat quality and featured producing of Meat enriched-Se.   however, I have minor comments that might be considered:

2. In the introduction section, please emphasis on the added value/novelty of this research as organic selenium was used as feed additives for  more 20 years with great success.

3. Here some references that could be of added value to your Ms:

-Hassan Fawzia, Samia Mobarez, Manal Mohamed, Y. A. Attia , Aml Mekawy and K. Mahrose (2021). Zinc and/or Selenium Enriched Spirulina as Antioxidants in Growing Rabbit Diets to Alleviate the Deleterious Impacts of Heat Stress during Summer Season. Animals 2021, 11, 756. https://doi.org/10.3390/ani11030756

-Attia, Y. A., A. A. Abdalah, Zeweil H. S., Bovera F., A. A. Tag El-Din, M. A. Araft (2010). Effect of inorganic or organic selenium supplementation on productive performance, egg quality and some physiological traits of dual purpose breeding hens.  Cezh J. Animal Science 55: 505–519.

4.  L 41-43, in the conclusion section, please add the dose of Se for taken home massage

5. L 93, Plz indicate the form of feed.

6. What is the number of samples used in the item 2.2 - 2.5, they must be stated? L 103-153.

7.  In the statistical section, plz provide the statistical model, experimental Unit, and check for if Tukey post hock is differences than Duncan for mean comparison for the interaction cells, Tukey is stronger than Duncan due to control of type 1 experimental error.

8. In the tables, the numbers and decimals in tables should be follow the rule of: xxxx, xxx, xx.x, x.xx, 0.xxx and 0.0xxx

9.  In the conclusion section, please add the dose of recommended Selenium.

Author Response

(The authors gave the same response as above.)

Reviewer 3 Report

The paper examines the effects of Cardamine violifolia, a Se-enriched plant, on plasma biochemical parameters, anti-oxidative capacity, intestinal morphology, and meat quality of broilers challenged with lipopolysaccharide. The topic is very relevant since there is a growing demand for poultry products of high quality and safety. The paper presents novel and useful findings. The introduction provides evidence-based background for the research. The methods have been properly described, the results are well presented, and the data interpretation is appropriate. The findings are thoroughly discussed, and conclusions are justified by the results. I did not find any major errors.

Comments are provided within the attached article.

I recommend acceptance of the article after minor corrections.

All the best

Author Response

(The authors gave the same response as above.)

Round 2

Reviewer 2 Report

The authors have done a good job  in the revised copy, but I noticed that reference 6 should be corrected  As Attia et al for the sequences of the authors.  

Author Response

Thank you for your comments. We have corrected the sequences of the authors.